# DomainFusion: Generalizing To Unseen Domains with Latent Diffusion Models

## Abstract

Latent diffusion model(LDM) has achieved success in various tasks beyond image generation due to its large-scale image-text training datasets and high-quality generation capability. However, its application in image classification remains unclear. Existing approaches directly transform LDM into discriminative models, which involve using mismatched text-image pairs that LDM fail to present accurate estimation, resulting in degraded performance. Other methods that extract vision knowledge are only designed for generative tasks. Additionally, domain generalization (DG) still faces challenges due to the scarcity of labeled cross-domain data. Existing data-generation approaches suffer from limited performance, and how to immigrate LDM to DG remains unknown. Therefore, we concern these two issues and propose a framework DomainFusion, which leverages LDM in both latent level and pixel level for DG classification. In latent level, we propose Gradient Score Distillation(GSD) which distills gradient priors from LDM to guide the optimization of the DG model. We further theoretically proved it can optimize the KL divergence between the predicted distributions of LDM and the DG model. In pixel level, we propose an autoregressive generation method to shuffle synthetic samples and a sampling strategy to optimize the semantic and non-semantic factors for synthetic samples. Experimental results demonstrate that DomainFusion surpasses data-generation methods a lot and achieves state-of-the-art performance on multiple benchmark datasets.

## 1 Introduction

Latent diffusion models have shown particular effectiveness in generating high-quality images through stable and scalable denoising objectives(Rombach et al., 2022). Thanks to large-scale image-text datasets(Schuhmann et al., 2022) and novel generative model architectures(Ho et al., 2020), latent diffusion models have also been demonstrated to encapsulate transferable vision knowledge, indicating their potential for other vision tasks. Consequently, recent research has focused on utilizing these rich transferable visual features for various visual perception tasks(Zhang et al., 2023), achieving successful applications in text-to-3D generation(Poole et al., 2022; Wang et al., 2023) and image editing(Hertz et al., 2023). However, as is shown in Figure 1, they can not be extended to discriminative tasks. For image classification tasks, how to leverage diffusion models for semantic understanding remains unclear. The main challenge lies in the unknown image categories, as categories represent the fundamental semantic descriptions of images, making it difficult to formulate appropriate text prompts for denoising and efficiently harness the multimodal understanding capability of latent diffusion models.

Hence, our primary focus is on leveraging latent diffusion models for image classification tasks especially in domain generalization(DG) scenario. DG is a particularly challenging image classification task, as it necessitates models to maintain high classification accuracy on unseen target domains. As shown in Figure 1, despite previous attempts such as diffusion classifiers(Li et al., 2023; Clark & Jaini, 2023), which matches image with all possible labels and gives score vectors based on the denosing loss, the effectiveness of such approaches is limited. As they directly transform the latent diffusion model into discriminative model(discriminative model is used interchangeably with image classification model throughout), matching the image with all category labels, which includes inaccurate estimations and leads to confusing results. We substantiate this claim in Figure 1 that latent diffusion can merely present accurate prediction on matched text-image pairs and fails

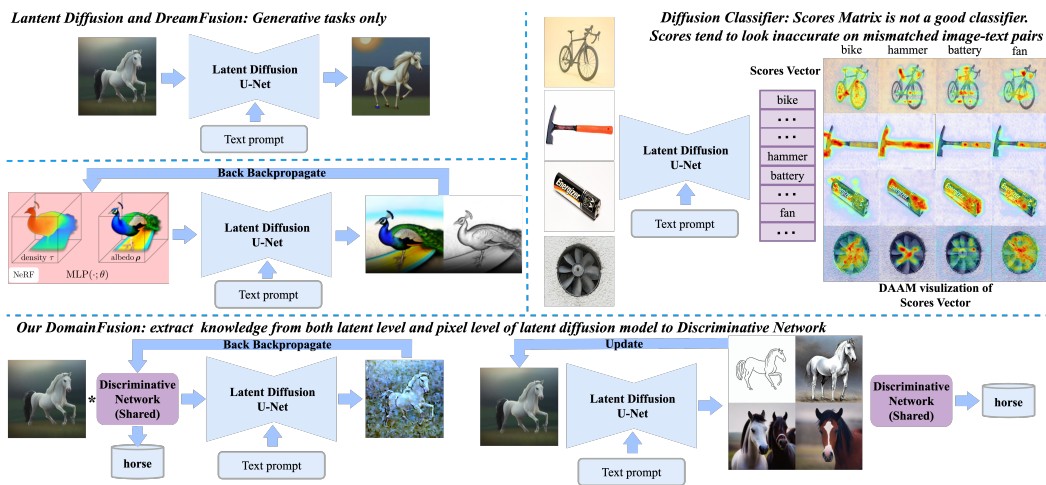

Figure 1: Comparison of DomainFusion with Latent Diffusion(Rombach et al., 2022), DreamFusion(Poole et al., 2022), and Diffusion Classifier(Li et al., 2023). Latent diffusion and DreamFusion are restricted in generative tasks. Diffusion Classifier performs poorly as a classifier due to directly use latent diffusion as discriminate model. We substantiate this conclusion by visualizing the score vector using cross-attention maps derived from DAAM(Tang et al., 2022). For example, an image of a hammer contains no information about a battery and latent diffusion fails to understand such pairs. In contrast, DomainFusion distills latent diffusion's vision knowledge by gradient prior and simultaneously utilizes latent diffusion's generation capacity to synthesize novel samples. Details are shown in Figure 3.

on mismatched scenarios, where we visualize the score vectors of these approaches using cross-attention maps obtained by DAAM(Tang et al., 2022). It can be observed that latent diffusion fails to comprehend mismatched image-text pairs, resulting in noisy score estimations. Intuitively, this limitation also holds true since, for example, as shown in Figure 1, an image of a hammer contains no information about a battery, thus the inaccurate estimation cannot be used as classification score.

Hence, it is imperative to leverage the transferable vision knowledge of latent diffusion while alleviating mismatched text-image pairs. This necessitates the extraction of visual representations from latent diffusion and distilling them into an additional network rather than directly using it as discriminated model. Based on existing research, there are two approaches for extracting transferable vision knowledge. The first approach involves replacing the encoder with a denoising U-Net to extract feature maps and cross-attention maps, followed by additional training of a downstream decoder(Zhao et al., 2023; Zhang et al., 2023). However, this approach assumes knowledge of the image categories, rendering it inapplicable to image classification tasks. The second approach, exemplified by Score Distillation sampling(Poole et al., 2022; Wang et al., 2023), introduces a loss function based on probability density distillation, utilizing a 2D latent diffusion model as a prior to optimize a parameterized 3D image generator. However, this new approach is limited to generative models and cannot be employed for discriminative tasks. To address this, we propose a novel approach Gradient Score Distillation(GSD), which for the first time distills vision knowledge from latent diffusion and applies it to domain generalization.

Domain generalization(DG) has advanced significantly but is still hindered by limited labeled cross-domain data(Wang et al., 2022). To address this, researchers have focused on generating diverse data to augment the source domain(Zhou et al., 2020b;a). However, these methods suffer from limited performance and how to immigrate latent diffusion models to DG remains unknown.

Therefore, we concern the aforementioned two issues, including the difficulty in extracting vision knowledge from latent diffusion for image classification, and the uncertainty of applying latent diffusion in domain generalization to achieve high performance, and propose DomainFusion, the first approach that leverages latent diffusion models for DG classification. In DomainFusion, we incorporate latent diffusion models in both latent-level and pixel-level, as shown in Figure 1 and Figure 3. In latent level, we propose Gradient Score Distillation (GSD), which establishes a connection between the parameter spaces of discriminative and latent diffusion, optimizing the former's parameter space using gradient prior derived from the latter. In pixel space, we adopt an autoregressive generation approach to continuously shuffle a synthetic dataset. To ensure that the generated samples are better suited for DG, we propose a sampling strategy. Specifically, we generate multiple candidate samples

and optimize the combination of semantic and non-semantic factors through a sampling strategy to synthesize the final new samples. Experimentally, our DomainFusion outperforms state-of-the-art methods in multiple benchmark datasets using multiple backbones, confirming the effectiveness of DomainFusion.

Our key contributions can be summarized as follows:

- To the best of our knowledge, we propose the first framework that leverages latent diffusion models in both latent level and pixel level for domain generalization, extracting the vision knowledge in latent diffusion models to facilitate the high-level comprehension.

- We propose Gradient Score Distillation(GSD) which leverages gradient priors from the latent diffusion model to guide the optimization process of the DG model. We presents theoretical proof of its effectiveness as optimizing the KL divergence between the predicted distributions of the latent diffusion model and the discriminative model, thereby providing supervised signals.

- We propose an autoregressive generation method to shuffle synthetic samples and a sampling strategy to optimize the semantic and non-semantic factors for synthetic samples.

## 2 RELATED WORK

### 2.1 DOMAIN GENERALIZATION

Most domain generalization (DG) methods operate under the assumption of having access to a sufficient amount of cross-domain data. The main focus of these methods is to eliminate domain-specific biases and retain invariant features across multiple source domains, including learning more generalized feature representations via domain-invariant representation learning(Krueger et al., 2021; Rosenfeld et al., 2020) and feature disentanglement(Wang et al., 2021; Zhang et al., 2022), optimization-based methods via meta-learning (Bui et al., 2021; Zhang et al., 2021)and ensemble learning(Li et al., 2022b; Arpit et al., 2022). Despite their success, these methods are limited by the scarcity of real-world cross-domain data, which hinders their practical applicability. As an alternative strategy, another line of research focuses on data augmentation to generate new domains and diverse samples(Zhao et al., 2021; Zhou et al., 2021; Li et al., 2021; 2022a). Our method integrates the advantages of both approaches, as we not only tackle the issue of data scarcity but also emphasize how to assist the model in learning more effective domain-invariant feature representations.

### 2.2 DIFFUSION MODELS FOR PERCEPTION VISION TASKS

Diffusion models have emerged as the state-of-the-art in image generation tasks(Nichol et al., 2021; Saharia et al., 2022; Ramesh et al., 2022; Rombach et al., 2022). Moreover, they have proven to be successful in various perception vision tasks, including image classification(Li et al., 2023; Clark & Jaini, 2023), image segmentation(Tan et al., 2022), object detection(Chen et al., 2022), monocular depth estimation(Zhao et al., 2023), and semantic correspondence(Zhang et al., 2023). Significantly, a substantial amount of research efforts has been dedicated to extracting valuable vision knowledge from diffusion models. In line with our objective of leveraging the diffusion model for domain generalization in image classification tasks, we classify existing approaches into three distinct groups based on their relevance to classification tasks and the methodologies of utilizing latent diffusion. The first group involves directly converting latent diffusion models from generative to discriminative tasks without training extra models, such as the aforementioned diffusion zero-shot classifier (Li et al., 2023; Clark & Jaini, 2023). While this approach provides valuable insights, it suffers from limited technical scalability, slow inference speed, and suboptimal performance. The second group focuses on extracting feature maps and cross-attention maps from the denoiser to train an extra decoder for downstream tasks(Zhang et al., 2023; Zhao et al., 2023). However, this approach often requires prior knowledge of image categories, which are used as conditional inputs into the denoising process. As a result, it is not suitable for high-level visual tasks like image classification. The third group are based on Score Distillation Sampling(SDS)(Poole et al., 2022; Wang et al., 2023; Hertz et al., 2023; Kim et al., 2023), which demonstrate good scalability. However, to the best of our knowledge, this approach is only applicable to generative models intending for generating diffusion-like images, which means the critical prerequisite is ensuring that the trained model shares the same image generation objectives with latent diffusion models. Hence, our primary focus lies

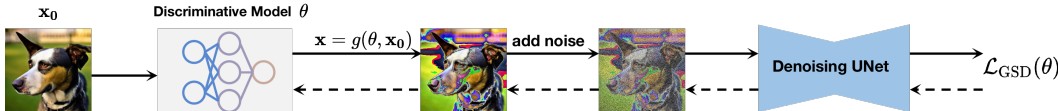

Figure 2: For our Gradient Score Distillation, we process a raw image $\mathbf{x}$ by $\mathbf{x} = g(\theta, \mathbf{x_0})$ via a discriminative model $\theta$, and then add noise to extract loss $\mathcal{L}_{\text{GSD}}(\theta)$ which allows the gradient flows from the latent diffusion model to the discriminative model's parametric space. As an example, we use a randomly initialized two-layer MLP here by having $g(\theta, \mathbf{x_0}) = \mathbf{x_0} \cdot \text{MLP}(\theta, \text{flatten}(\mathbf{x_0}))$.

in exploring **leveraging high-level semantic knowledge from latent diffusion models in a more natural manner to optimize discriminative models**.

## 3 METHOD

In this section, we first introduce diffusion preliminaries briefly. Then we elaborate DomainFusion from Gradient Score Distillation in section 3.2. Besides, we give a brief comparison from Proxy A-distance(PAD) perspective to show its effectiveness. and theoretically proved that GSD minimizes the KL divergence between the predicted distributions of the latent diffusion and the discriminative model, thereby providing supervised signals. Then we elaborate autoregressive generation and sampling method in section 3.3, and the overall loss extraction framework ins section 3.4.

### 3.1 DIFFUSION PRELIMINARIES

**A Recap of Diffusion Model.** Diffusion models are latent variable generative models defined by a forward and reverse Markov process(Ho et al., 2020). In the forward process $\{q_t\}_{t \in [0,T]}$, Gaussian noise is progressively added into the data $\mathbf{x}_0 \sim q_0(\mathbf{x}_0)$. The forward process at timestep $t$ is expressed as $q(\mathbf{x}_t \mid \mathbf{x}_0) = \mathcal{N}(\mathbf{x}_t; \alpha_t \mathbf{x}_0, \sigma_t^2 \mathbf{I})$, where $\sigma_t$ and $\alpha_t^2 = 1 - \sigma_t^2$ are hyperparameters satisfying $\sigma_0 \approx 0$ and $\sigma_1 \approx 1$. In the reverse process $\{p_t\}_{t \in [0,T]}$, Gaussian noise is progressively removed by an optimal MSE denoiser(Sohl-Dickstein et al., 2015) $\hat{\boldsymbol{\epsilon}}_\phi(\boldsymbol{x}_t, t)$ from $p(\mathbf{x}_T) = \mathcal{N}(\mathbf{0}, \mathbf{I})$ to reconstruct the clean data $\mathbf{x}_0$, which is typically given by transitions $p_\phi(\mathbf{x}_{t-1} \mid \mathbf{x}_t) = \mathcal{N}(\mathbf{x}_{t-1}; \mathbf{x}_t - \hat{\boldsymbol{\epsilon}}_\phi(\mathbf{x}_t; t), \sigma_t^2 \mathbf{I})$. The optimal MSE denoiser $\hat{\boldsymbol{\epsilon}}_\phi(\boldsymbol{x}_t, t)$ is trained with a weighting function $w(t)$ that varies with the timestep $t$ by minimizing:

$$\mathcal{L}_{\text{Diff}}(\phi; \mathbf{x}) = \mathbb{E}_{t \sim \mathcal{U}(0,1), \boldsymbol{\epsilon} \sim \mathcal{N}(\mathbf{0}, \mathbf{I})} \left[ w(t) \| \hat{\boldsymbol{\epsilon}}_\phi(\alpha_t \mathbf{x}_0 + \alpha_t \boldsymbol{\epsilon}; t) - \boldsymbol{\epsilon} \|_2^2 \right] \tag{1}$$

**Diffusion Models as a Potential Generative Classifier.** For image classification tasks, the fundamental requirement is to compute the log-likelihood over class labels $\{y_i\}$. Unfortunately, diffusion models do not produce exact log-likelihoods(i.e. directly computing $\log p_\phi(\boldsymbol{x} \mid y = \mathrm{y}_i)$ is intractable)(Ho et al., 2020). However, recent research(Li et al., 2023; Clark & Jaini, 2023) has provided compelling evidence that $\log p_\phi(\boldsymbol{x} \mid y = \mathrm{y}_i)$ can be estimated using $\mathcal{L}_{\text{Diff}}$. This is attributed to a profound interrelation between the log-likelihood over its variational lower bound (ELBO) and $\mathcal{L}_{\text{Diff}}$. Specifically, the relationship can be articulated as follows:

$$\log p_\phi(\mathbf{x}_0 \mid y) \geq \text{ELBO} \approx -\mathbb{E}_{t, \boldsymbol{\epsilon}} \left[ w(t) \| \hat{\boldsymbol{\epsilon}}_\phi(\mathbf{x}_t; y; t) - \boldsymbol{\epsilon} \|_2^2 \right] + C = -\mathcal{L}_{\text{Diff}} + C \tag{2}$$

where $C$ is a constant independent of the class labels $y$. Therefore, $-\mathcal{L}_{\text{Diff}}$ **can be employed as a proxy to estimate the log-likelihood over class labels** $\log p_\phi(\mathbf{x}_0 \mid y)$(Li et al., 2023; Clark & Jaini, 2023), thereby transforming diffusion models into a potential classifier.

### 3.2 LEVERAGING LATENT SPACE BY GRADIENT SCORE DISTILLATION

Existing research on leveraging semantic knowledge within latent diffusion models can be categorized into three approaches. However, each approach has its limitations as analysed in Section 2.2. Our main focus is to explore a more natural and efficient way of utilizing the high-level semantic knowledge from latent diffusion models to optimize discriminative models for image classification. Therefore, we propose GSD, which utilizes gradient prior from latent diffusion to optimize the parameter space of discriminative model by backpropagation.

Given the discriminative network $\theta$ to be trained and an image $x_0$ with its class label $y_0$ (note that we employ $y$ throughout to represent both the numeric class label within the discriminative network $\theta$

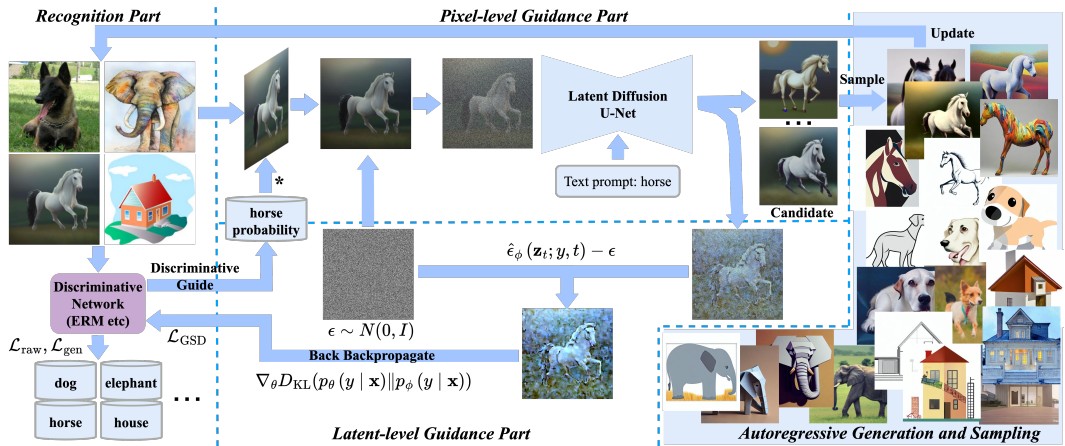

Figure 3: DomainFusion can be divided into four components. In Recognition Part, we start with the original source dataset and the synthetic dataset generated later for supervised training. In Latent-level Guidance Part, each image $\mathbf{x}$ is fed into the DG Discriminative Network $\theta$ and weighted by the output confidence w.r.t. its label $y$. $\mathbf{x}$ is then added noise $\boldsymbol{\epsilon}$ and passed through the latent diffusion U-Net $\phi$ along with $y$ as text prompt, resulting in predicted noise $\hat{\boldsymbol{\epsilon}}_\phi(\mathbf{x}_t; y; t)$. The discrepancy between the predicted and real noise $[\hat{\boldsymbol{\epsilon}}_\phi(\mathbf{x}_t; y; t) - \boldsymbol{\epsilon}]$ is utilized to obtain $\mathcal{L}_{\text{GSD}}$, which updates the parameter space of $\theta$ by backpropagation. In Pixel-level Guidance part, DomainFusion decodes the U-Net denosing result into multiple candidates. In Autoregressive Generation and Sampling Part, the sampling strategy is utilized to optimize the combination of semantic and non-semantic factors of candidates and ultimately sample one novel image to update $\mathbf{x}$ in Recognition Part.

and the textual class label within the latent diffusion model $\phi$), our initial step involves forwarding $x_0$ through $\theta$ for image classification, and compute the element-wise product of $x_0$ with the confidence score corresponding to class $y_0$ to obtain a $\theta$-related image $x$. We denote this "pseudo-generative" process as $\mathbf{x} = g(\theta) = p_\theta(y \mid \mathbf{x}_0)\,\delta(y_0)\,\mathbf{x}_0$. Subsequently, we feed $\mathbf{x}$ into the denoising process of $\phi$), and we denote the loss generated from denoising as $\mathcal{L}_{\text{GSD}}$. Note that $\mathcal{L}_{\text{GSD}}$ is mathematically equivalent to $\mathcal{L}_{\text{Diff}}$ in Equation 1, which yields:

$$\mathcal{L}_{\text{GSD}}(\phi; \mathbf{x} = g(\theta)) = \mathcal{L}_{\text{Diff}}(\phi; \mathbf{x}) = \mathbb{E}_{t,\boldsymbol{\epsilon}}\left[w(t)\,\|\hat{\boldsymbol{\epsilon}}_\phi(\alpha_t\mathbf{x} + \alpha_t\boldsymbol{\epsilon}; t) - \boldsymbol{\epsilon}\|_2^2\right] \tag{3}$$

We compute the gradient of $\mathcal{L}_{\text{GSD}}$ w.r.t. $\theta$ while omitting the U-Net Jacobian term following the SDS setting(Poole et al., 2022):

$$\nabla_\theta \mathcal{L}_{\text{GSD}}(\phi, \mathbf{x} = g(\theta)) = \mathbb{E}_{t,\epsilon}\left[w(t)\,(\hat{\epsilon}_\phi(\mathbf{z}_t; y, t) - \epsilon)\,\frac{\partial \mathbf{x}}{\partial \theta}\right] \tag{4}$$

$\nabla_\theta \mathcal{L}_{\text{GSD}}$ is then used to update $\theta$ through backward propagation. Through this approach we establish a pathway for gradient propagation from the latent diffusion models to the discriminative models. In the subsequent analysis, we shall delve deeper into the properties of GSD concerning its role in enhancing transferable semantic understanding.

**Gradient Score Distillation facilitates high-level visual learning.** To verify the effectiveness of GSD, we compute the Proxy A-distance (PAD)(Ding et al., 2022). PAD requires extracting image features separately from the source and target domains, labeling them as 1 and 0, and subsequently training a classifier to discriminate between these two domains. Given a test error of $\varepsilon$, PAD is defined as $2(1 - 2\varepsilon)$. A superior DG algorithm yields a lower PAD, indicating its ability to extract domain-invariant features. Consistent with prior studies(Ding et al., 2022; Glorot et al., 2011; Chen et al., 2012; Ajakan et al., 2014), we employ DomainFusion with/without GSD to extract image features from source and target domains, labeled as 1 and 0, and train a linear SVM for classification. As is shown in Figure 4, we first quantify PAD between a single source domain and the target domain, demonstrating that incorporating GSD consistently yields lower PAD values. Then we measure the PAD between all source domains and target domain, revealing a larger margin between the two versions, thus validating the effectiveness of GSD.

We now delve into a theoretical explanation of why GSD facilitates the learning of high-level semantic knowledge in discriminative models. Given a $\mathbf{x}$ and label $y$, $\theta$ produces the log-likelihood $p_\theta(y \mid \mathbf{x})$. Conversely, the latent diffusion model also inherently estimates the log-likelihood

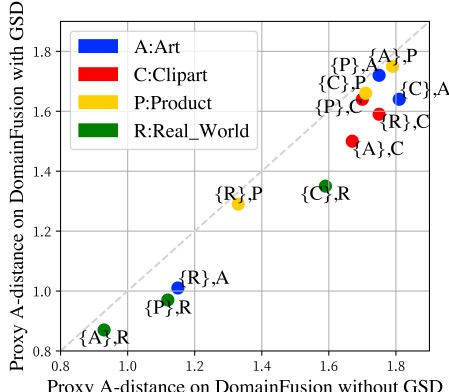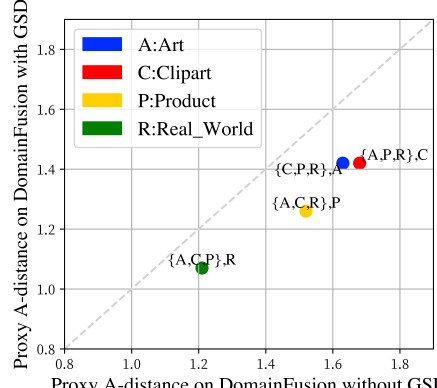

Figure 4: Proxy A-distance (PAD) on Office Home. x axis: PAD computed upon DomainFusion without GSD; y axis: PAD computed upon DomainFusion with GSD. We employ the DG model to extract features across diverse domains to train a linear domain classifier, and PAD is proportional to its classification accuracy. A superior DG model yields a lower PAD, indicating its ability to extract domain-invariant features. DomainFusion with GSD demonstrates lower PAD compared to its non-GSD version in both cases: (a) PAD between single source domain and the target domain. For example, {C},A denotes using the model to measure PAD between one source domain Clipart and the target domain Art. (b) PAD between all source domains and the target domain.

$p_\phi\left(y \mid \mathbf{x}\right)$. We compute the KL divergence between them and its gradient w.r.t. $\theta$ by:

$$\nabla_\theta D_{\mathrm{KL}}(p_\theta\left(y \mid \mathbf{x}\right) \| p_\phi\left(y \mid \mathbf{x}\right)) = \mathbb{E}_{p_\theta}\left[\nabla_\theta \log p_\theta\left(y \mid \mathbf{x}\right)\right] - \mathbb{E}_{p_\theta}\left[\nabla_\theta \log p_\phi\left(y \mid \mathbf{x}\right)\right] \quad (5)$$

Based on the conclusion derived from Equation 2, 3, we employ $-\mathcal{L}_{\mathrm{GSD}}$ to estimate $\log p_\phi\left(\mathbf{x} \mid y\right)$:

$$\nabla_\theta D_{\mathrm{KL}}(p_\theta\left(y \mid \mathbf{x}\right) \| p_\phi\left(y \mid \mathbf{x}\right)) = \mathbb{E}_{p_\theta}\left[\nabla_\theta \log p_\theta\left(y \mid \mathbf{x}\right)\right] + \mathbb{E}_{p_\theta}\left[\nabla_\theta \mathcal{L}_{\mathrm{GSD}}\right] \quad (6)$$

As a consequence of incorporating an impulse function $\delta$ into the computation of $\mathcal{L}_{\mathrm{GSD}}$, the resulting value of $\mathcal{L}_{\mathrm{GSD}}$ becomes a fixed constant when calculating $\mathbb{E}_{p_\theta}\left[\nabla_\theta \mathcal{L}_{\mathrm{GSD}}\right]$:

$$\mathbb{E}_{p_\theta}\left[\nabla_\theta \mathcal{L}_{\mathrm{GSD}}\right] = \int \nabla_\theta \mathcal{L}_{\mathrm{GSD}} \cdot p_\theta\left(y\right) dy = \nabla_\theta \mathcal{L}_{\mathrm{GSD}} \int p_\theta\left(y\right) dy = \lambda \nabla_\theta \mathcal{L}_{\mathrm{GSD}} \quad (7)$$

Where $\lambda$ is a constant unrelated to $\theta$. Despite the integral value should be 1 due to the presence of softmax, certain constants were omitted during the derivation from Equation 3 to Equation 4. Consequently, $\lambda$ is introduced here to rectify this omission. Since the computation of the second term (B) is independent of that of the first term (A), we can minimize the KL divergence by:

$$\min \nabla_\theta D_{\mathrm{KL}}(p_\theta\left(y \mid \mathbf{x}\right) \| p_\phi\left(y \mid \mathbf{x}\right)) = \min \underbrace{\mathbb{E}_{p_\theta}\left[\nabla_\theta \log p_\theta\left(y \mid \mathbf{x}\right)\right]}_{(A)} + \min \underbrace{\lambda \nabla_\theta \mathcal{L}_{\mathrm{GSD}}}_{(B)} \quad (8)$$

Within our DomainFusion the framework (elaborated in Chapter 3.2), we employ explicit minimization of the term (B) and simultaneously utilize the images involved in $\mathcal{L}_{\mathrm{GSD}}$ computation for supervised training, thereby concomitantly minimizing the term (A). **Consequently, we effectively minimize the KL divergence between the two distributions.** Therefore, GSD can be expressed as distilling high-level semantic knowledge from the latent diffusion models and utilizing it as a supervisory signal, thereby facilitating the comprehension of high-level semantic features. This approach introduces a new paradigm for leveraging visual representations from latent diffusion models.

### 3.3 LEVERAGING PIXEL SPACE BY AUTOREGRESSIVE GENERATION AND SAMPLING

**Autoregressive Generation to augment source domain.** Synthetic images have demonstrated their potential in augmenting the source domain to facilitate models in learning more generalized feature representations(Wang et al., 2022), thereby enhancing the model's DG performance. Nevertheless, the notion that larger synthetic datasets equate to superior performance does not hold true. Research has indicated that when training exclusively on synthetic data without the inclusion of real data, there is a notable decline in model performance(Bansal & Grover, 2023; Azizi et al., 2023), which can be attributed to domain shift. Therefore, it is necessary to leverage the efficient generation capability of latent diffusion models while preventing the synthetic dataset from taking over the real dataset. To address this, we maintains a synthetic dataset at a 1:1 ratio in size with the training data. Besides, we propose an autoregressive approach that allows for dynamic updates to the latent diffusion model's

input. This approach continuously shuffles the diversity of synthesized dataset, thereby fostering improved DG performance.

**Sampling to optimize semantic and non-Semantic factor combination.** To mitigate domain shift, existing research has provided valuable insights that synthetic data for DG should possess semantic factors similar and introduce different non-semantic factors(Dai et al., 2023)(e.g., content and style). Drawing from this intuition, we propose a sampling mechanism to optimize the combination of semantic factors and non-semantic factors for generated images. Specifically, given an input image $\mathbf{x}_0$ and $N$ generated candidate samples $\{\mathbf{x}_i\}_{i\in[1,N]}$, we decompose them into semantic factors (content) and non-semantic factors (style), represented as $\{c_i\}_{i\in[0,N]}$ and $\{s_i\}_{i\in[1,N]}$. To select the most diverse non-semantic factor $s^*$, we apply the KL divergence to measure their distance w.r.t. $s_0$ by $s^* = \arg\max_{s_i} KL(\mu_i, \mu_0) + KL(\sigma_i, \sigma_0)$, where $\mu$ and $\sigma$ represent the mean and variance components of style $s$ respectively. Additionally, for selecting the most similar semantic factor, we employ both the cosine similarity and the $\theta$) classification confidence by $c^* = \arg\max_{c_i} \lambda cos(f_\theta(c_i), f_\theta(c_0)) + (1 - \lambda)p_\theta(y|c_i)$, where $f_\theta$ represents the feature map extracted by $\theta$ and $\lambda$ represents a predetermined constant. Subsequently, we utilize AdaIN style transfer(Huang & Belongie, 2017) to sample the ultimate new sample $\mathbf{x}^*$ by $\mathbf{x}^* = \sigma^* c^* + \mu^*$.

### 3.4 LOSS EXTRACTION AT BOTH LATENT AND PIXEL LEVELS

Finally, we arrive at the DomainFusion algorithm. The overall training architecture is:

$$\mathcal{L} = \lambda_1\mathcal{L}_{\text{raw}} + \lambda_2\mathcal{L}_{\text{gen}} + \lambda_3\mathcal{L}_{\text{GSD}} \tag{9}$$

where $\mathcal{L}_{\text{raw}}$ and $\mathcal{L}_{\text{gen}}$ denote the cross-entropy loss in the source dataset and the synthesized dataset respectively, $\lambda_1,\lambda_2$ and $\lambda_3$ are predetermined hyper-parameters. By employing this method, we effectively extract loss from both the latent level and pixel level of the latent diffusion model, thereby achieving the first comprehensive solution for DG utilizing the latent diffusion model.

## 4 EXPERIMENTS

### 4.1 EXPERIMENTAL SETTINGS

**Settings and Datasets.** Following DomainBed(Gulrajani & Lopez-Paz, 2020), we conducted a series of experiments on five prominent real-world benchmark datasets: PACS(Li et al., 2017), VLCS(Fang et al., 2013), OfficeHome(Venkateswara et al., 2017), TerraIncognita(Beery et al., 2018), and DomainNet(Peng et al., 2019). To ensure a fair and consistent comparison, we follow DomainBed's training and evaluation protocol. We provide full details in Appendix A.2.

**Implementation Details.** For the latent diffusion model, we employ the stable diffusion v1-4 model card. The batch size is set to 16 and we employ the Adam optimizer(Kinga et al., 2015) and cosine learning rate schedule. We provide full details in Appendix A.2.

### 4.2 MAIN RESULTS

**Comparison with domain generalization methods.** We compare DomainFusion with baseline methods and recent DG algorithms and present results in Table 1. In the first section, we evaluated DomainFusion using the ResNet-50(He et al., 2016) architecture as the backbone. The experimental results demonstrate that DomainFusion exhibits a significant lead w.r.t. other generation-involved methods by +1.8pp, +1.3pp, +4.1pp, +3.2pp, and +5.2pp in PACS, VLCS, Office Home, Terrainc, and DomainNet, respectively. Moreover, DomainFusion also outperforms the current state-of-the-art methods in all benchmark datasets, yielding accuracy improvements of +0.8pp, +0.2pp, +1.7pp, +0.7pp, and +0.3pp in each dataset.

In the second section of Table 1, we employ Regnet-Y-16GF(Radosavovic et al., 2020) as the backbone and utilize the SWAG(Singh et al., 2022) method to obtain a pre-trained model on the ImageNet(Russakovsky et al., 2015) dataset, aiming to investigate the maximum performance potential of the DomainFusion algorithm. The experimental results convincingly demonstrate a significant performance improvement exhibited by the DomainFusion algorithm compared to ERM across all

Table 1: Comparison with DG methods.The DG accuracy on five domain generalization benchmarks are presented with the best results highlighted in bold. The results denoted by † correspond to the reported numbers from DomainBed(Gulrajani & Lopez-Paz, 2020). Results of other DG methods including Fish(Shi et al., 2021), SelfReg(Kim et al., 2021), mDSDI(Bui et al., 2021), MIRO(Cha et al., 2022), Fishr(Rame et al., 2022) are from corresponding paper. And results of Diffusion Classifier(Li et al., 2023) and DiffusionNet(Clark & Jaini, 2023) are implemented by us.

| Algorithm | PACS | VLCS | OfficeHome | TerraInc | DomainNet | Avg. |
|---|---|---|---|---|---|---|
| *Diffusion-based image classification method* | | | | | | |
| Diffusion Classifier | 47.0 | 40.6 | 26.8 | 13.5 | 10.8 | 27.7 |
| DiffusionNet | 23.8 | 0.8 | 15.5 | 8.5 | 0.3 | 9.8 |
| *Using ResNet-50 backbone: Non-generation method* | | | | | | |
| ERM† | 85.5 | 77.5 | 66.5 | 46.1 | 40.9 | 63.3 |
| MLDG† | 84.9 | 77.2 | 66.8 | 47.7 | 41.2 | 63.6 |
| CORAL† | 86.2 | 78.8 | 68.7 | 47.6 | 41.5 | 64.5 |
| MMD† | 84.7 | 77.5 | 66.3 | 42.2 | 23.4 | 58.8 |
| DANN† | 83.6 | 78.6 | 65.9 | 46.7 | 38.3 | 62.6 |
| MTL† | 84.6 | 77.2 | 66.4 | 45.6 | 40.6 | 62.9 |
| SagNet† | 86.3 | 77.8 | 68.1 | 48.6 | 40.3 | 64.2 |
| RSC† | 85.2 | 77.1 | 65.5 | 46.6 | 38.9 | 62.7 |
| Fish | 85.5 | 77.8 | 68.6 | 45.1 | 42.7 | 63.9 |
| SelfReg | 85.6 | 77.8 | 67.9 | 47.0 | 42.8 | 64.2 |
| mDSDI | 86.2 | 79.0 | 69.2 | 48.1 | 42.8 | 65.1 |
| MIRO | 85.4 | 79.0 | 70.5 | 50.4 | 44.3 | 65.9 |
| Fishr | 85.5 | 77.8 | 68.6 | 47.4 | 41.7 | 64.2 |
| *Using ResNet-50 backbone: Generation-involved method* | | | | | | |
| GroupDRO† | 84.4 | 76.7 | 66.0 | 43.2 | 33.3 | 60.7 |
| Mixup† | 84.6 | 77.4 | 68.1 | 47.9 | 39.2 | 63.4 |
| Mixstyle‡ | 85.2 | 77.9 | 60.4 | 44.0 | 34.0 | 60.3 |
| **DomainFusion(ours)** | **87.0** | **79.2** | **72.2** | **51.1** | **44.6** | **66.8** |
| *Using RegNetY-16GF backbone with SWAG pre-training* | | | | | | |
| ERM | 89.6 | 78.6 | 71.9 | 51.4 | 48.5 | 68.0 |
| MIRO | **97.4** | 79.9 | 80.4 | 58.9 | 53.8 | 74.1 |
| **DomainFusion(ours)** | 96.6 | **80.0** | **83.4** | **60.6** | **55.9** | **75.3** |

datasets. Moreover, our proposed approach outperforms the current SOTA algorithm, MIRO(Cha et al., 2022), in all datasets except PACS, with performance gains of +0.1pp, +3pp, +1.7pp, and +2.1pp in VLCS, OfficeHome, TerraInc, and DomainNet, respectively. Our algorithm's effectiveness has been substantiated through a wide range of experiments.

**Comparison with other diffusion-based image classification methods.** There are two existing methods that employ latent diffusion models for image classification, we denote them as Diffusion Classifier(Li et al., 2023) and DiffusionNet (Clark & Jaini, 2023) respectively. Therefore, we also conduct experiments to compare DomainFusion with them. To ensure fairness, we constrain all three approaches to use stable diffusion and employ the same latent diffusion model parameters, including a fixed image size of 320*320 and unified text prompt template.The experimental results are presented in Table 1. The findings indicate that both the Diffusion Classifier and DiffusionNet do not exhibit high performance as DG image classifiers.

## 4.3 ABLATION STUDY

We conduct experiments on Office Home for ablation study based on RegNet-Y-16GF.

**Effects of Different Components.** As shown in Table 1, $\mathcal{L}_{\text{gen}}$ improves the average accuracy by 4.9% by generating a more diverse set of samples to augment the source domain, resulting in a significant improvement in DG performance. However, using $\mathcal{L}_{\text{gen}}$ alone still exhibits a considerable performance gap compared to state-of-the-art methods. To address this discrepancy, $\mathcal{L}_{\text{GSD}}$ bridges this gap by further enhancing the accuracy by 4.8% compared to use $\mathcal{L}_{\text{gen}}$ solely.

Table 2: Effects of Different Components in DomainFusion

| $\mathcal{L}_{\text{raw}}$ | $\mathcal{L}_{\text{gen}}$ | $\mathcal{L}_{\text{GSD}}$ | Art | Clipart | Product | Real | Avg. |
|:---:|:---:|:---:|:---:|:---:|:---:|:---:|:---:|
| ✓ | ✗ | ✗ | 69.3 | 61.3 | 81.6 | 82.5 | 73.7 |
| ✓ | ✓ | ✗ | 73.6 | 71.2 | 80.7 | 88.7 | 78.6 |
| ✓ | ✓ | ✓ | **81.2** | **73.9** | **88.5** | **90.1** | **83.4** |

**Effects of the Sampling Strategy.** We also provide ablation study on the effect of the sampling strategy, and details can be seen in Appendix A.3.

**Effects of the Candidate Number.** Moreover, we provide ablation study on the impact of the number of candidates. Details can be seen in Appendix A.3.

## 4.4 VISUALIZATION

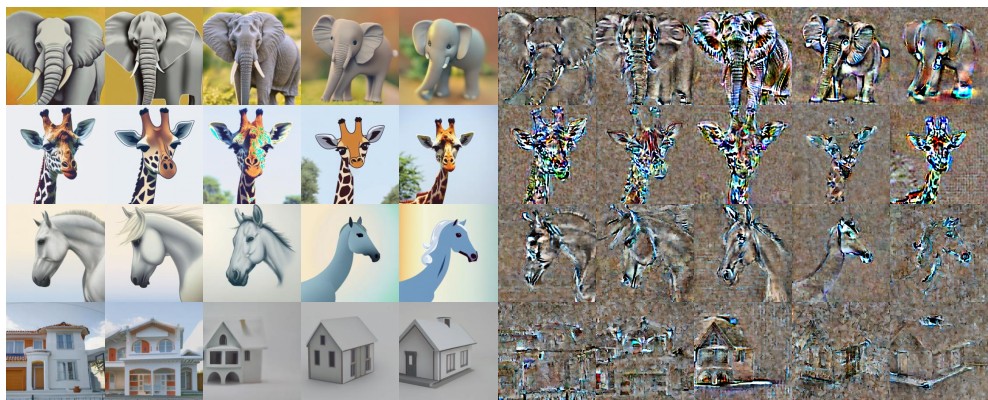

Figure 5: Visualization of generated samples and GSD noise, with the left section being autoregressively generated samples and the right section being corresponding GSD noise.

**Visualization of generated samples.** Figure 5 showcases the visualization results of the global synthetic dataset at various iterations. As we employed an autoregressive generation approach, with each row representing the iterative evolution of a specific image. In terms of visual effects, it is apparent that as the synthesized dataset is updated, the image sequences retain a certain degree of semantic similarity, while also introducing new non-semantic features. This serves as evidence of the effectiveness of our method.

**Visualization of GSD noise.** Figures 5 illustrates the visualization of the GSD noise for all images. We first calculate the difference between predicted denoising latent and latent with noise, and then use the stable diffusion decoder to decode this difference. It is evident that these noise patterns effectively capture the high-level semantic information in the images while reducing the influence of irrelevant elements, such as the background. This finding demonstrates the strong generalization capability of the latent diffusion model, as it can extract transferable feature representations, which contribute to optimizing the DG semantic understanding network in our GSD, further confirming the effectiveness of the GSD method.

## 5 CONCLUSION

In this paper , we propose the first framework which utilizes the latent diffusion model (LDM) in both the latent level and pixel level for domain generalization (DG) classification. In latent level, we propose Gradient Score Distillation (GSD) that extracts transferable knowledge as gradient priors from the LDM to optimize the DG model. The effectiveness of GSD is theoretically proved as optimizing the KL divergence between the predicted distributions of the LDM and the DG model. In pixel level, we propose an autoregressive generation method to continuously shuffle synthetic samples and a sampling strategy to optimize the combination of semantic and non-semantic factors in synthetic samples. Experimental results demonstrate that our method achieves state-of-the-art performance on DG classification.

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

## A  APPENDIX

### A.1  Comparison with Score Distillation Sampling and Diffusion classifier

**Comparison with Score Distillation Sampling.** (1) GSD extends SDS paradigm to discriminative models and perception tasks. SDS is inherently restricted to cases where the targeted model is a generative model, thus limiting its applicability to tasks beyond generation. In contrast, Our GSD employs diffusion-like images as intermediaries to establish a connection between the parameter spaces of discriminative and latent diffusion models, facilitating the transfer of semantic knowledge for discriminative tasks. (2) GSD provides clearer evidence of its effectiveness. Equation 8 yields a compelling conclusion that GSD can be employed to optimize the KL divergence between the prediction distributions of the latent diffusion model and the DG network. This implies that GSD can provide supervision signals for the DG network similar to ground truth in supervised learning.

**Comparison with Diffusion classifier.** Empirically and experimentally, we find that the diffusion classifier(Li et al., 2023; Clark & Jaini, 2023) that directly uses noise for classification does not yield satisfactory results. The key reason behind this is that the diffusion classifier requires matching the correct image with a fake category and predicting the probability of this fake match. For example, using a picture of a dog and the text promt 'cat', the diffusion classifier is expected to provide the probability of the dog picture belonging to the cat category. However, the dog picture does not contain any information about cats. Consequently, utilizing incorrectly matched image-text pairs leads to noisy and inaccurate predictions. In contrast, our GSD approach **merely utilizes correctly matched image-text pairs**, effectively eliminating noisy predictions. Figure 1 illustrates the results of visualizing the diffusion classifier's score vectors by cross-attention map in the UNet obtained by DAAM(Tang et al., 2022). Images from Office Home are sequentially matched with a real label prompt and fake label prompts to compute the cross-attention map. It can be observed that diffusion fails to comprehend mismatched image-text pairs, resulting in unreliable predictions in such cases.

### A.2  Experimental Settings

**Settings and Datasets.** Following DomainBed, we conducted a series of experiments on five prominent real-world benchmark datasets: PACS(4 domains, 9,991 samples, and 7 classes), VLCS(4 domains, 10,729 samples, and 5 classes), OfficeHome(4 domains, 15,588 samples, and 65 classes), TerraIncognita(4 domains, 24,778 samples, and 10 classes), and DomainNet(6 domains, and 586,575 samples, and 345 classes). To ensure a fair and consistent comparison, we follow DomainBed's(Gulrajani & Lopez-Paz, 2020) established training and evaluation protocol. In this protocol, we designate one domain as the target, while the remaining domains serve as source domains. Model selection is conducted using the training-domain validation approach, where 20% of the source domain data is used for validation. The performance of domain generalization is evaluated individually on each domain and then averaged across all domains.

**Implementation Details.** For the latent diffusion model, we employ the stable diffusion v1-4 model card. Specifically, we utilize the image-to-image pipeline for image generation and loss extraction, where the input image size is set to 320x320, which greatly boosts algorithm training speed and reduces computational overhead, and other hyperparameters are set to their default values as specified by stable diffusion. For domain generalization, we utilize ResNet-50 pretrained on ImageNet and RegNet-Y-16GF pretrained using SWAG as our backbone models. The batch size is set to 16, except for DomainNet where it is reduced to 8 due to computational limitations. We employ the Adam optimizer and cosine learning rate schedule during training.

### A.3  Ablation study

We conduct experiments on Office Home for ablation study. All models are based on RegNet-Y-16GF and trained for 120 epochs.

**Effects of Different Components.** As shown in Table 1, $\mathcal{L}_{\text{gen}}$ improves the average accuracy by 4.9% by generating a more diverse set of samples to augment the source domain, resulting in a significant improvement in DG performance . However, using $\mathcal{L}_{\text{gen}}$ alone still exhibits a considerable performance gap compared to state-of-the-art methods. To address this discrepancy, $\mathcal{L}_{\text{GSD}}$ bridges this gap by further enhancing the accuracy by 4.8% compared to use $\mathcal{L}_{\text{gen}}$ solely.

Table 1: Effects of Different Components in DomainFusion

| $\mathcal{L}_{\text{raw}}$ | $\mathcal{L}_{\text{gen}}$ | $\mathcal{L}_{\text{GSD}}$ | Art | Clipart | Product | Real | Avg. |
|---|---|---|---|---|---|---|---|
| ✓ | ✗ | ✗ | 69.3 | 61.3 | 81.6 | 82.5 | 73.7 |
| ✓ | ✓ | ✗ | 73.6 | 71.2 | 80.7 | 88.7 | 78.6 |
| ✓ | ✓ | ✓ | **81.2** | **73.9** | **88.5** | **90.1** | **83.4** |

Table 2: Effects of the Sampling Strategy.

| w/o | Art | Clipart | Product | Real | Avg. |
|---|---|---|---|---|---|
| ✗ | 79.4 | 71.8 | 87.5 | 88.2 | 81.7 |
| ✓ | **81.2** | **73.9** | **88.5** | **90.1** | **83.4** |

Table 3: Effects of the Candidate Number.

| candidate number | Art | Clipart | Product | Real | Avg. |
|---|---|---|---|---|---|
| $N = 1$ | 79.4 | 71.8 | 87.5 | 88.2 | 81.7 |
| $N = 2$ | **81.2** | **73.9** | **88.5** | **90.1** | **83.4** |
| $N = 5$ | 80.4 | 72.7 | 87.8 | 89.3 | 82.6 |

Table 4: Time cost hours of different components.

| Algorithm | Clipart | Info | Painting | Quickdraw | Real | Sketch | Avg. |
|---|---|---|---|---|---|---|---|
| Diffusion Classifier | 5.8 | 6.3 | 8.9 | 20.3 | 20.7 | 8.3 | 11.7 |
| DomainFusion without GSD | 21.1 | 17.4 | 18.5 | 17.8 | 17.4 | 17.2 | 18.2 |
| DomainFusion with GSD | 28.2 | 25.2 | 26.0 | 25.4 | 25.2 | 25.2 | 25.9 |

**Effects of the Sampling Strategy.** Table 2 demonstrates the effect of the sampling strategy. The inclusion of the sampling strategy led to a significant enhancement of 1.7% in accuracy compared to the exclusion version, thereby indicating the effectiveness of the sampling strategy. The implementation of the sampling strategy allows for the optimization of both semantic and non-semantic factors, resulting in the generation of samples that are better aligned with the requirements of DG.

**Effects of the Candidate Number.** Table 3 presents the impact of the number of candidates, denoted as $N$, on the results. We considered three scenarios: $N = 1$, $N = 2$, and $N = 5$, with $N = 2$ being the default setting for DomainFusion. In the implementation process, $N$ is primarily adjusted by the number of images generated for each prompt in the stable diffusion pipeline. It is noteworthy that a larger value of $N$ may yield a decline performance because too many candidates may lead to visual clutter in the synthesized images. Therefore, setting $N$ as 2 is deemed as a favorable choice.

## A.4 COST ANALYSIS

We analyze the GPU time consumption of different components in DomainFusion on DomainNet, along with the runtime of the Diffusion Classifier for comparison. It is worth noting that all the reported times refer to the number of hours the algorithms consumed on 8* V100 GPUs. DomainFusion was run for 120 epochs and completed both training and inference, while the Diffusion Classifier only completed the inference phase. Despite the longer runtime of DomainFusion compared to the Diffusion Classifier, it remains affordable while achieving a significant improvement in accuracy. Note that when used for inference, our DomainFusion requires no extra time compared with ERM.

## A.5 MORE VISUALIZATION RESULTS

We present more visualization results of autoregressively generated samples and corresponding GSD noise images in Figure 1.

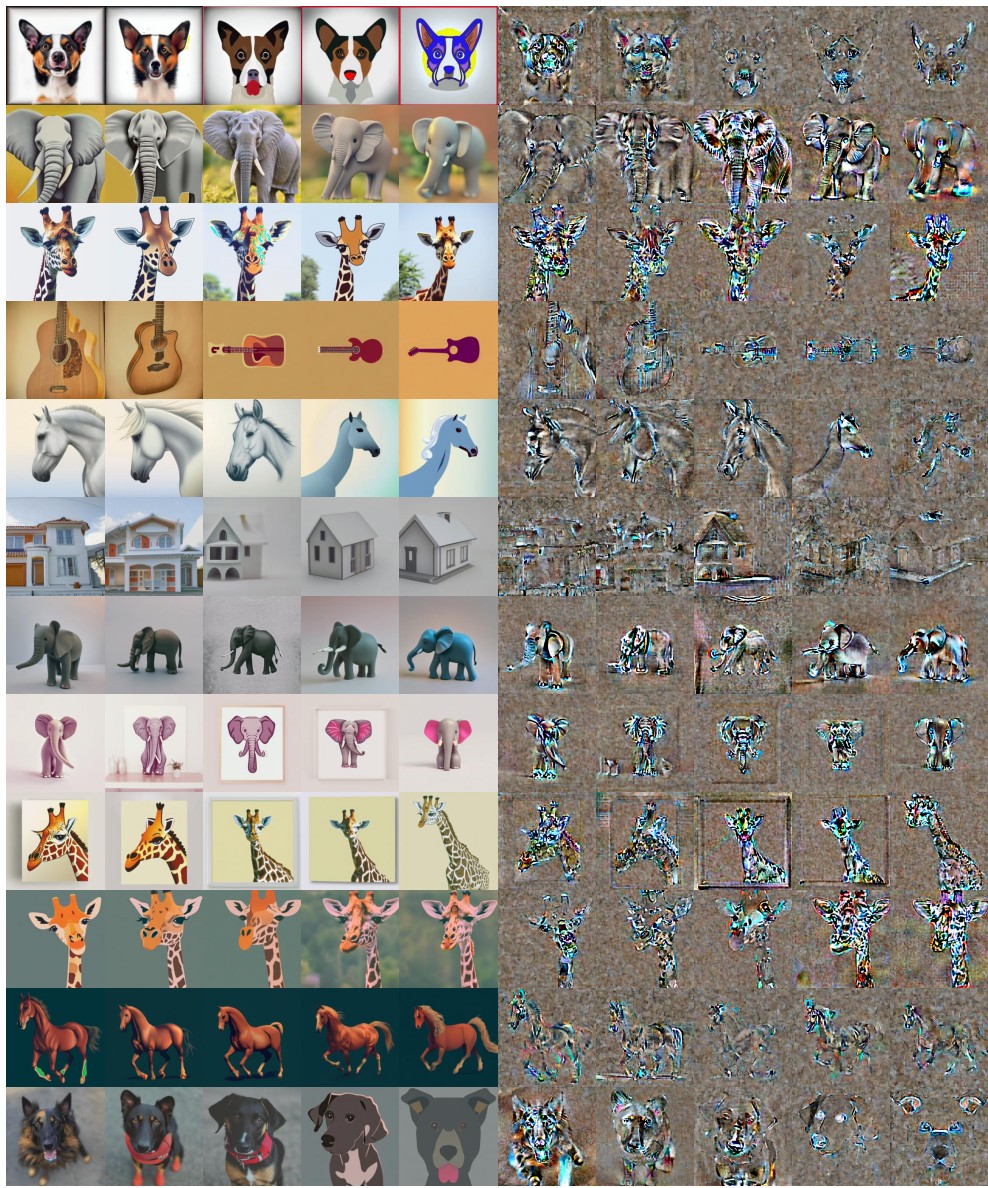

Figure 1: More visualization results of generated samples and GSD noise, with the left section being autoregressively generated samples and the right section being corresponding GSD noise.

