# OpenReview forum: "DomainFusion: Generalizing To Unseen Domains with Latent Diffusion Models"
_ICLR.cc/2024/Conference — ICLR 2024 Conference Withdrawn Submission_

### Official Review · Reviewer_JYFK · 2023-10-26

**Soundness:** 3 good
**Presentation:** 3 good
**Contribution:** 2 fair
**Rating:** 5
**Confidence:** 4

**Summary:**

This paper investigates how to utilize a pre-trained latent diffusion model to assist in domain generalization tasks. Specifically, the author employs diffusion in two ways. First, the author introduces a gradient score distillation loss to distill gradient priors from LDM to the classification model, which is achieved through the denoising objective. Secondly, the LDM is applied to synthetic data, and the paper proposes a method to utilize synthetic data via style decomposition. Experiments are conducted on standard DG benchmarks.

**Strengths:**

1. This is the first paper I have seen that employs the denoising loss to assist with DG classification, and the use of this denoising loss is theoretically explained as minimizing the distance between two distributions.
2. While using augmented or synthetic data for DG has been widely studied, this paper indeed introduces a novel approach to leverage synthetic data via style decomposition and validates its superiority over vanilla methods.
3. The proposed method is validated on standard benchmark datasets.

**Weaknesses:**

1. To be frank, the GSD loss proposed in this paper is essentially the adoption of the denoising loss from diffusion. In other words, the loss from diffusion is directly used here. Although the authors provide a theoretical explanation, stating that this approach can minimize the distance between two distributions, this mainly leverages the merits of diffusion rather than being a unique contribution of this paper.
2. Moreover, I'm unclear why the classification probability is element-wise multiplied with the input image. I understand that this can facilitate gradient flow from the diffusion model to the classification model. However, this multiplication remains somewhat perplexing. For instance, during training, the classification model can easily fit the training data, making the classification probability for the true label approach 1. In such a scenario, does this multiplication retain its significance? Or is there a straightforward interpretation of the physical meaning of the gradient back-propagated to the "horse probability" in Fig3? Does it aim to increase or decrease the horse probability?
3. Regarding the augmentation sampling loss, I find it reasonable. However, it lacks a comparison with existing augmentation methods. For instance, the current method identifies the most similar semantic and most diverse non-semantic. While experiments show its advantage over direct image generation, a comparison with existing augmentation techniques is expected. Methods like MixStyle [1] and EFDM [2] have achieved improved results through mixing styles, so I would like to see a comparison with such methods, perhaps by using MixStyle/EFDM on different x_i as augmented images.
4. The writing could be improved: a) at the end of Page 8, "as shown in Table 1" should be refined to "as shown in Table 2". b) What does "Tab 4" in the appendix specifically refer to? Does it indicate the runtime of different methods? c) On Page 7, the sentence "we employ both the cosine similarity and the \theta)" should be refined.

Minor questions:
Why does the result of L_{raw} only in Table 2 (73.9) differ from the ERM result (71.9) in Table 1? They should use the cross-entropy loss only.

[1] Domain Generalization with MixStyle
[2] Exact Feature Distribution Matching for Arbitrary Style Transfer and Domain Generalization

**Questions:**

See weaknesses.

---

### Official Review · Reviewer_4xQn · 2023-10-30

**Soundness:** 3 good
**Presentation:** 2 fair
**Contribution:** 2 fair
**Rating:** 5
**Confidence:** 3

**Summary:**

The paper proposes a framework that leverages LDMs for domain generalization in image classification tasks. The authors try to address the challenge of utilizing LDMs for discriminative tasks and the scarcity of labeled cross-domain data in DG. They propose two components: Gradient Score Distillation (GSD) at the latent level and autoregressive generation with a sampling strategy at the pixel level. The experimental results show that DomainFusion outperforms existing methods on multiple benchmark datasets.

**Strengths:**

1. The paper targets an important problem in utilizing LDMs for image classification and DG. The proposed framework provides a comprehensive solution that leverages LDMs at both the latent and pixel levels.
2. A GSD component that distills gradient priors from LDMs to guide the optimization of the DG model.
3. The autoregressive generation method and sampling strategy contribute to improving the diversity and quality of synthetic samples for DG.

**Weaknesses:**

- The presentation of the paper is poor and hard to follow. The motivation is not well explained. In Fig. 1, various methods are in a mess, making it difficult to understand what the paper is trying to do. I suggest the author carefully consider the organization of the introduction and the content in Fig. 1.

- Besides, it seems that the paper does not under careful proof-reading and there are many typos in it, e.g., “denosing”->“denoising” and "they directly transform the latent diffusion model into **a** discriminative model" in the intro section.

- For the part of the experiment, the paper lacks of comparison with the latest SOTA methods. Besides, it seems that the improvement is not very significant when taking the training cost into consideration (8x V100).

- Besides, I am also in doubt about the performance of the model under non-natural image settings, e.g., among medical images. Since using LDM is not a strictly fair comparison due to the additional usage of the prior distribution of the natural image.

**Questions:**

Please see the weakeness part

---

### Official Review · Reviewer_CwAu · 2023-10-30

**Soundness:** 3 good
**Presentation:** 3 good
**Contribution:** 3 good
**Rating:** 5
**Confidence:** 4

**Summary:**

In this paper, the authors incorporate the semantic information from LDM to the domain generalization model through the proposed Gradient Score Distillation (GSD). Moreover, an autoregressive generation method is proposed to shuffle synthetic samples and a sampling strategy is used to optimize the semantic and non-semantic factors for synthetic samples. Experiments show that the method works very well on different datasets.

**Strengths:**

- The idea is pretty interesting to provide high level semantic information from the generative model to the discriminative one, and the proposed GSD works well to help improve the performance through ablation study.
- The visualization of the mismatch between image and text can help better understanding the subject.

**Weaknesses:**

- Equation (5) seems problematic:
$$D_{KL}(p_\theta(y|x)|| p_\phi(y|x))=E_{p_\theta}(\log p_\theta(y|x)) - E_{p_\theta}(\log q_\phi(y|x))$$
$$\nabla_\theta D_{KL}(p_\theta(y|x)|| p_\phi(y|x))=\nabla_\theta E_{p_\theta}(\log p_\theta(y|x)) - \nabla_\theta E_{p_\theta}(\log q_\phi(y|x))$$
$$\nabla_\theta E_{p_\theta}(\log  p_\theta(y|x)) = \nabla_\theta \int  p_\theta(y|x) \log  p_\theta(y|x)dy = \int \nabla_\theta (p_\theta(y|x) \log  p_\theta(y|x)) \neq E_{p_\theta}(\nabla_\theta \log p_\theta(y|x))$$
We can get similar result for the second part $\nabla_\theta E_{p_\theta}(\log q_\phi(y|x))$. Therefore, equation (5) does not hold.

- Sec 3.3 is not very clear and it's better to add more details to better illustrate the idea. For example,
     - Does the authors mean to dynamically replace some samples in the synthetic dataset with fixed number of samples during the training, or obtain the dataset before the training of the DG model?
    - How to compute $\{c_i\}$ and $\{s_i\}$?

- It's better to rephrase equation (4), where $\frac{\partial x}{\partial \theta}$ should be the Jacobian matrix and other part is a vector, and it is a matrix-vector multiplication form.

**Questions:**

- What's the output $y$ of the discriminative model, is it a scalar or vector?
- How to use it to reweight the input $x_0$?
- Why using $x=g(\theta, x_0)$ in Fig.2  as the input of LDM instead of $x_0$?

---

### Official Review · Reviewer_MNgv · 2023-11-01

**Soundness:** 1 poor
**Presentation:** 2 fair
**Contribution:** 2 fair
**Rating:** 3
**Confidence:** 4

**Summary:**

This paper proposed an approach of leveraging pretrained latent diffusion models for the domain generalization classification task. They derive a training objective inspired by SDS of DreamFusion, and combine this training objective with synthetic data augmentation in the pixel space. With these two techniques, they got reasonably well performance on the task.

**Strengths:**

- It is worthwhile to explore the potential of diffusion models in terms of representation learning, and in general the problem of leveraging diffusion models for extracting high-level semantic representations, such as classification, hasn't been fully solved.
- Extensive baselines approaches have been compared to show the effectiveness of the approach.

**Weaknesses:**

My main concern is that the derivation of the training objective is not valid, with a lot of errors:
  - Eq. 4: why could you drop the Jacobian term? The DreamFusion training objective can be justified from the perspective of variational inference + sticking the landing. But it is not well-justified in the case of this paper.
  - Eq. 4 is not intuitive either. In fact, $\epsilon$ equal 0 in expectation, so the training objective effectively is optimizing $x$ such that the resulting $z_t$ minimizes $\hat{\epsilon}$, equivalently maximizing the score at time step $t$. However, the way $x$ being parametrized is just an observed data $x_0$ multiplied by a scalar scaling factor $p_\theta(y_0|x_0)$. Why would we expect learning a scaling factor of $x_0$ that maximizes the score function can give meaningful training signal?
  - I don't get the rational to parametrize $x = p(y_0|x_0) x_0$ either. Looks like it just parametrized in this way such that it is easier to derive something close to SDS.
  - If I understand correctly, $z_t$ in eq. 4 is derived by adding noise to the scaled clean data $x$. However, since $x$ is being scaled by $p_\theta(y_0|x_0)$, which could be out of distribution of what $\hat{\epsilon}$ was trained on, how do we make sure $\hat{\epsilon}$ is still trustworthy?
  - Eq. 5: you cannot move $\nabla_\theta$ inside the expectation directly, as the expectation is taken over $p_\theta$ which is dependent on $\theta$. The correct way is first using re-parametrization trick introduced by VAE to make the expectation taken over something that is not dependent on $\theta$.
  - Eq. 8: for (A) term, it is not equivalent to the cross-entropy or supervised learning objective, because both $x$ and $p_\theta$ contain parameter $\theta$. Based on chain rule, it should be $(\log p_\theta)' \cdot g(\theta)' = (\log p_\theta)' \cdot x_0 \cdot (p_\theta)'$.

I don't find a detailed explanation about the autoregressive generation approach in the main text, which is claimed to be one of main contributions, making it hard to evaluate this contribution. The separation of semantic and non-semantic factors is also not clear.

**Questions:**

Please see comments above.